# Economic and Clinical Benefits of Bivalent Respiratory Syncytial Virus Prefusion F (RSVpreF) Maternal Vaccine for Prevention of RSV in Infants: A Cost-Effectiveness Analysis for Mexico

**DOI:** 10.3390/vaccines13010077

**Published:** 2025-01-16

**Authors:** José Luis Huerta, Robyn Kendall, Luka Ivkovic, Carlos Molina, Amy W. Law, Diana Mendes

**Affiliations:** 1Pfizer, Mexico City 05120, Mexico; joseluis.huerta@pfizer.com (J.L.H.); carlosandres.molina@pfizer.com (C.M.); 2Evidinno Outcomes Research, Vancouver, BC V5Y 1K2, Canada; rkendall@evidinno.com (R.K.); livkovic@evidinno.com (L.I.); 3Pfizer Inc., New York, NY 10001, USA; amy.law@pfizer.com; 4Pfizer Ltd., Tadworth KT20 7NY, UK

**Keywords:** cost-effectiveness, respiratory syncytial virus (RSV), RSVpreF vaccine, maternal immunization, Mexico, economic evaluation, modeling

## Abstract

Background/Objectives: Respiratory syncytial virus (RSV) is a leading cause of respiratory infections in children. A novel RSVpreF vaccine for use among pregnant women for the prevention of RSV in infants is expected to be licensed in Mexico. Hence, the clinical and economic burden of RSV among infants in Mexico, with and without a year-round RSVpreF maternal vaccination program, was estimated. Methods: A cohort model was developed to project clinical and economic outcomes of RSV from birth to 1 year of age for maternal vaccination and no intervention. Incremental cost-effectiveness ratios were calculated from direct cost outcomes, life years, and quality-adjusted life years (QALYs). The value per dose of the RSVpreF for which the program would be cost-effective was explored. Analyses were conducted from the healthcare system perspective, with direct costs (2024 Mexican Pesos [MXN]) and outcomes discounted at 5% annually; scenario and sensitivity analyses tested the robustness of model settings and inputs. Results: Compared to no intervention, a year-round RSVpreF vaccine administered to 1891 M pregnant women would prevent 15,768 hospitalizations, 5505 emergency department cases, and 5505 physician office visits annually, averting MXN 1754 M in direct medical costs with an increase of 3402 life years or 3666 QALYs. The RSVpreF vaccine would be cost-saving up to MXN 1301/dose and cost-effective up to MXN 2105–MXN 3715/dose under an assumed cost-effectiveness threshold range of 1–3× the gross domestic product (GDP) per capita (MXN 247,310) per QALY gained. Conclusions: Year-round RSVpreF maternal vaccination would substantially reduce RSV’s clinical and economic burden among infants in Mexico and likely be a cost-effective program.

## 1. Introduction

Respiratory syncytial virus (RSV) is a respiratory virus that causes mild, cold-like symptoms and is the leading cause of lower respiratory tract infections (LRTIs) (e.g., bronchiolitis and pneumonia) in children younger than one year of age [1]. Worldwide, RSV-LRTI has an associated cost of EUR 4.82 billion, causing more than 100,000 deaths and over 3.5 million hospitalizations per year, primarily among infants in low- and middle-income countries (LMICs) [2]. Globally, approximately 97% of deaths due to RSV occur in LMICs and 39% of all hospital admissions due to RSV occur in infants less than six months of age [3].

In Mexico, according to data from Instituto Nacional de Estadística y Geografía (INEGI), among the total of 37 RSV deaths reported in 2022, 19 were infant deaths within the first year of life [4]. A recent burden of disease study quantified medical costs incurred among pediatric patients under 2 years of age diagnosed with RSV and risk factors from the perspective of Mexico’s public health institutions [5]. The study reported that the annual cost for RSV infections estimated by severity and complications were between USD 691.25 and USD 3677.91 for upper respiratory tract infections (URTIs), USD 247.83 and USD 10,508.66 for bronchiolitis, and USD 453.33 and USD 18,128.08 for pneumonia [5]. Mild complications largely contributed towards indirect costs, comprising 25% to 30% of the total cost, and represented 3 to 5 days of work absence. Furthermore, moderate and severe complications associated with RSV significantly influenced the total economic impact due to costs associated with hospitalizations.

A novel bivalent RSV prefusion F (RSVpreF) protein-based vaccine for the prevention of RSV (specifically, the RSV A and RSV B antigenic subgroups) in infants is expected to be licensed in Mexico for active vaccination of pregnant women between 24 and 36 weeks of gestational age (wGA). The RSVpreF vaccine has been approved by the Food and Drug Administration (FDA) [6] and European Medicines Agency (EMA) [7] based on data from the pivotal Phase 3 clinical trial (MATernal Immunization Study for Safety and Efficacy [MATISSE]), a randomized, double-blinded, placebo-controlled study which evaluated the efficacy and safety of maternal RSVpreF vaccination in preventing RSV-associated LRTI in infants [8]. In the MATISSE study, the efficacy of the RSVpreF vaccine against severe medically attended RSV-LRTI was 81.8% (99.5% confidence interval [CI] 40.6–96.3) within the first 3 months (90 days) of life, and 69.4% (97.6% CI 44.3–84.1) over the following 6 months [8]. For medically attended RSV-LRTI, RSVpreF vaccine efficacy was 57.1% (99.5% CI 128 14.7–79.8) through the first 3 months (90 days) of life, and 51.3% (97.6% CI 29.4–66.8) over the first 6 months [8].

The objective of this study is to perform a cost-effectiveness analysis from the Mexican healthcare system perspective to estimate the clinical and economic burden of RSV-LRTI among infants in Mexico, with and without a year-round RSVpreF maternal vaccination program.

## 2. Materials and Methods

A population-based Markov hypothetical cohort model was developed to estimate clinical and economic outcomes of RSV from birth to one year of age for RSVpreF vaccine compared to no intervention (Figure 1). The model population was defined based on wGA at birth. Preceding model entry, it was assumed that infants were either protected against RSV due to maternal vaccination or received no intervention. Expected clinical outcomes were projected for the model population monthly based on age, wGA at birth, intervention status, time since receipt of interventions (accounting for variation in the timing of receipt of interventions and rates of RSV during a one-year period), RSV disease rates, and mortality rates. Clinical outcomes included RSV cases stratified by care setting (i.e., hospital, emergency department [ED], or outpatient), deaths associated with RSV treated in-hospital, life years, and quality-adjusted life years (QALYs) for RSVpreF vaccine compared to no intervention. Economic outcomes included vaccine costs (RSVpreF vaccine and administration), medical costs associated with infants, and indirect costs for caregiver workdays lost and travel expenses (for scenario analysis). The model was adapted from a model presented at the Advisory Committee on Immunization Practices (ACIP) meeting in September 2023 [9] and was calibrated to reflect Mexican healthcare settings.

The base case analysis reflected the Mexican healthcare system perspective and included only intervention and direct medical costs (2024 Mexican Pesos [MXN]). A cost-effectiveness threshold of 1–3× gross domestic product per capita (GDPpc) was assumed (MXN 247,310–MXN 741,930) per QALY gained [10]. The model was adapted with a lifetime time horizon and a 5% annual discount rate was applied for costs and outcomes.

### 2.1. Population

The number of pregnant women giving birth in a given calendar year (1,891,388) and the total number of infants born (1,916,429), accounting for multiple births (e.g., twins, triplets) and stillbirths in a given calendar year, as well as the distribution of births over calendar months, were based on data from the Instituto Nacional de Estadística y Geografía (INEGI) for 2022 [11,12,13]. Liveborn infants were categorized by term status according to the following distribution: 91.8% born at ≥37 wGA (full term), 7.2% born at 32–36 wGA (late preterm), 0.7% born at 28–31 wGA (early preterm), and 0.3% born at ≤27 wGA (extreme preterm) [14]. Stillborn infants were defined similarly: 0.2% born at ≥37 wGA (full term), 2.2% born at 32–36 wGA (late preterm), 13.6% born at 28–31 wGA (early preterm), and 74.3% born at ≤27 wGA (extreme preterm) [4].

### 2.2. Epidemiology

Age-specific annual rates of RSV encounters were estimated from recently published RSV rates for Mexico by Mata-Moreno et al. (2024) (Table 1) [15]. All hospitalized cases, emergency department visits, and outpatient visits of RSV were assumed to manifest as LRTIs based on the ICD-10 hospitalization codes included in Mata-Moreno et al. (2024) [15]. The proportion of RSV cases encountered in hospital and outpatient settings in Mexico reported by Gamiño-Arroyo et al. (2016) [16] was applied to the incidence of RSV hospital encounters from Mata-Moreno et al. (2024) [15] to derive the RSV outpatient and emergency department incidence rates (assumed equivalent to outpatient incidence [3]). Age-specific relative rates of RSV by birth term status were based on a study which reported cases by gestational age at birth and chronologic age at infection (Appendix A) [17]. The distribution of RSV encounters by calendar month was based on the published literature from Mexico in 2024 (Appendix A) [15].

### 2.3. Mortality

The infant mortality rate was obtained from the data published by INEGI for deaths in children under one year of age for 2022 (Appendix A) [18]. The mortality rates were adjusted through application of relative risk of death by wGA subgroup at birth, estimated based on Centers for Disease Control and Prevention (CDC) WONDER data (Appendix A) and used to calculate the mortality of the model population [19]. Case fatality rate (1.176 deaths per 100 hospitalizations) for RSV-associated in-hospital mortality in Mexico was derived from Mata-Moreno et al. (2024) [15]. The relative risk of death due to RSV hospitalization by gestational age at birth was assumed to vary according to published data [3,20,21]; lacking robust evidence to suggest otherwise, these data were assumed to be invariant by subgroup of interest and by age (Appendix A).

### 2.4. Vaccine Effectiveness and Strategy

Vaccine effectiveness was based on “MATISSE” clinical trial efficacy data against medically attended RSV-LRTI and medically attended severe RSV-LRTI [8]. MATISSE study methodology included the analysis of two seasons in both hemispheres, where respiratory viruses could circulate and differ between the south and north hemisphere. Waning of vaccine effectiveness by month of age was estimated using linear extrapolation of data for the primary and secondary trial endpoints up to six months, followed by an assumption of linear waning to 0% vaccine effectiveness by age nine months (Figure 2) [8]. The vaccine effectiveness among infants born < 32 wGA was conservatively assumed to be 0%, although serology data from MATISSE shows that infants achieve sufficient levels of neutralizing antibody titers provided there are 14 days between vaccination and delivery [22]; therefore, there will likely be protection among early preterm infants (i.e., 28–<32 wGA) born beyond 2 weeks after maternal vaccination to 6 months of age [23].

Maternal vaccine uptake was assumed to be 60% based on values reported for maternal tetanus/pertussis (Tdap) vaccination in Mexico [24]. Uptake was assumed to occur year-round and be invariant by calendar month of expected delivery. In the absence of Mexico-specific data, the distribution of vaccine uptake by wGA was based on Tdap vaccine data among pregnant women in Argentina (Appendix A) [25]. It was assumed that the distribution of uptake for RSVpreF vaccine followed the same pattern as the Tdap vaccine from 24 to 36 wGA (which is the administration period during which the maternal RSVpreF vaccine is expected to be indicated).

### 2.5. Utilities

Due to the unavailability of reliable utility estimates for healthy infants, it was assumed that the utility value would be 1 for infants without RSV. Utility values for the general population over 1 year of age (i.e., for calculating the ongoing impact of RSV) were sourced from the literature (Appendix A) [26]. Disutility values for infants and for their caregivers due to RSV were derived from the published literature (Appendix A) [27,28]. These values were used in the model to estimate QALYs (based on life years achieved with the applied utility value according to age), as well as future lost QALYs in the case of premature infant death due to RSV.

### 2.6. Costs

Costs were considered and reported in 2024 MXN. Costs for healthcare items were obtained from publicly available local sources and inflated to 2024 values using the Consumer Price Index data obtained from INEGI [29], where necessary. For costs derived from non-Mexican sources, cost values were converted to MXN using the historical Purchasing Power Parity (PPP) indexes available from the Organization for Economic Co-operation and Development (OECD) [30] and inflated accordingly.

#### 2.6.1. Direct Costs

Hospitalization-specific costs for infants < 6 months of age hospitalized with RSV were calculated by weighted average to account for those infants admitted to an intensive care unit (ICU) (8.05% [31] for 4.2 days [32] at a unit cost of MXN 74,584 per day based on Instituto Mexicano de Seguro Social [IMSS] data) [33] and then transferred to the general ward (for the remaining 3.1 days [31] at a unit cost of MXN 12,926 per day) [33], and those infants admitted to a general ward only. For infants 6–12 months of age hospitalized with RSV, a weighted average was calculated to account for those infants admitted to ICU (3.75% [31] for 4.2 days [32] at a unit cost of MXN 74,584 per day based on IMSS data) [33] and then transferred to the general ward (for the remaining 0.1 days [31] at a unit cost of MXN 12,926 per day) [33], and those infants admitted to a general ward only. The estimated cost of RSV hospitalization was MXN 115,206 and MXN 65,293 per episode for infants aged < 6 months and 6–12 months of age, respectively.

Emergency department (ED)-specific costs (MXN 4203) of RSV were based on Level-3 urgent care costs from IMSS data [33]. Outpatient-specific costs (MXN 1174) of RSV were based on Level-2 outpatient family physician consultation costs from IMSS data [33]. No wGA-specific estimates were available for the aforementioned data; therefore, no variation was assumed.

Direct costs also included costs associated with intervention (vaccine and administration costs) as well as medical care. The administration cost for maternal vaccination (MXN 255.49) was based on Government of Mexico data [34]. As the unit cost of the RSVpreF vaccine had not yet been determined in Mexico, we explored the value per dose of RSVpreF, for which the program would be cost-saving and cost-effective. Cost-effectiveness was determined assuming that the willingness-to-pay (WTP) threshold per QALY gained in Mexico would range between 1× and 3× GDPpc (MXN 247,310–MXN 741,930).

#### 2.6.2. Indirect Costs

Indirect costs comprising caregiver lost productivity and travel expenses were considered as a scenario analysis. Productivity losses were incorporated into the analysis in multiple forms: (1) work loss among caregivers due to caring for infected infants, (2) potential lost labor opportunities following the maturity of children who had died from RSV, and (3) costs associated with travel per RSV episode. Caregiver work losses were estimated by calculating the number of days missed at work and multiplying it to the average daily wage. The percentage of infants with their primary caregiver in the workforce (97.60%) and the average daily wage for full-time employed caregivers (MXN 148.40) were based on a 2023 estimate from Secretaría de Economía [35]. The percentage of caregivers with full-time employment (83%) was based on OECD data [36]. The number of days of parental care (5 days) in all settings (hospitalization, ED, and outpatient) were derived from a Brazilian study by Sartori at al. (2016) [37]. For future productivity loss, general (adult) population workforce participation and average wage by age were assumed based on INEGI data [38,39,40]. Travel cost per episode of RSV (MXN 122.68) was based on the published literature from El Salvador and Panama (Mexico-specific data were not available) [41]. Since no setting-specific data were available, no variation was assumed in travel cost per episode of RSV between hospitalization, ED, and outpatient settings.

### 2.7. Sensitivity Analysis

To assess the robustness of the results, one-way sensitivity analysis (OWSA) and probabilistic sensitivity analysis (PSA) were conducted. OWSA was performed by varying disease incidence, general infant mortality rate, case fatality rate due to RSV, vaccine effectiveness, cost of interventions, direct cost of disease, healthy infant utility and disutility, and caregiver QALY by ±25%. PSA was also conducted by random sampling (1000 simulations) from specified probability distributions assigned to input parameter values.

### 2.8. Scenario Analysis

Scenario analyses were conducted to determine the impact of alternative populations, assumptions, and model settings on the base case model results. Alternative scenarios assuming seasonal vaccination strategy and vaccine coverage of 40% and 80% were considered. Given the likely underestimation of the RSV-related case fatality rate in Mexico [15], alternative scenarios were considered for the case fatality rate, assuming 0.7 and 0.4 deaths per 100 hospitalizations for infants less than 6 months old and infants between 6 and 12 months old, respectively, from the global systematic literature review report [3] and 1.7 per 100 hospitalizations from Argentina [42]. Additionally, alternative scenarios for variable discount rates for costs (3% and 7%) and outcomes (0% and 7%), and the inclusion of indirect costs from a societal perspective, were considered.

## 3. Results

### 3.1. Base Case Analysis

Total and incremental costs and outcomes are shown in Table 2. The model estimated that incorporation of a year-round maternal RSVpreF vaccination program (assuming 60% coverage) would prevent 33% of hospital admissions (*n* = 15,768), 23% of emergency room visits (*n* = 5505), and 23% of outpatient visits (*n* = 5505) each year. Additionally, maternal vaccination with RSVpreF would prevent 31% of RSV-related deaths (*n* = 171) per year. Compared to no intervention, vaccination with RSVpreF would result in 3402 additional life years and 3666 QALYs gained, corresponding to savings of MXN 1754 million in direct medical care costs.

From a healthcare system perspective, the maternal RSVpreF vaccine would be cost-saving up to MXN 1301/dose and cost-effective up to MXN 2105-MXN 3715/dose, assuming a cost-effectiveness threshold range of 1–3×GDPpc per QALY gained in Mexico. Assuming a WTP of 1× GDPpc per QALY gained, the total incremental cost associated with maternal vaccination was MXN 905.84 million (Table 2).

### 3.2. One-Way Sensitivity Analysis

The resulting ICER values from the OWSA are presented in Figure 3 and Table 3. The parameters with the greatest influence on the ICER results included the effectiveness of RSVpreF vaccine, incidence of RSV-related hospitalization, cost of RSVpreF vaccine, cost of RSV-related hospitalization, and case fatality rate due to RSV-related hospitalization.

### 3.3. Probabilistic Sensitivity Analysis

The acceptability curves showed that the probability for the maternal vaccine being cost-effective at a WTP of 1–3×GDPpc per QALY gained was 64% to 93% compared to no vaccine (Figure 4).

### 3.4. Scenario Analysis

The results of the different scenario analyses are described in Table 4. When lower discount rates for future costs (3%) and no discounting of future benefits were modeled, the ICER of RSVpreF was 28.1% of 1×GDPpc; however, when both costs and outcomes were discounted at 7% annually, the ICER was 131% of 1×GDPpc. Adopting the societal perspective would reduce the ICER to almost 0.9 times 1×GDPpc. When the RSV vaccination strategy was changed from year-round to seasonal vaccination, the ICER of RSVpreF was 42.7% of 1×GDPpc. In the scenarios of lower (40%) and higher (80%) vaccine uptakes, the ICER of RSVpreF was 99.9% of 1×GDPpc. When lower case fatality rates (0.7 and 0.4 deaths per 100 hospitalizations) for infants less than 6 months old and infants between 6 and 12 months old, respectively, were modeled, the ICER was increased to 1.5 times 1×GDPpc; however, when higher case fatality rates were modeled (1.7 deaths per 100 hospitalizations), the ICER of RSVpreF was 72.1% of 1×GDPpc.

## 4. Discussion

With expected approval of a novel bivalent RSVpreF vaccine for pregnant women in Mexico, an economic evaluation was undertaken to examine the clinical and economic burden of RSV-LRTI among infants in Mexico, with and without a year-round RSVpreF maternal vaccination program. The results of this analysis showed that the maternal RSV vaccine is cost-effective compared to no vaccination strategy from both the Mexican healthcare system and societal perspectives. The findings suggested that year-round RSVpreF maternal vaccination would substantially reduce RSV’s clinical and economic burden among infants in Mexico and likely be a cost-saving program. Year-round RSVpreF administered to 1.9 million pregnant women would prevent 15,768 hospitalizations, 5505 ED cases, and 5505 outpatient visits. With 3666 QALYs gained, RSVpreF averts MXN 1754 million in direct medical costs. RSVpreF would be cost-saving up to MXN 1301/dose and cost-effective up to MXN 2105–3715/dose.

The maternal RSV vaccine remained a cost-effective strategy across all scenario analyses with the exception of the discount rate (7% for costs and outcomes). Compared with no vaccination strategy, the maternal RSV vaccine resulted in cost-savings for the scenario where seasonal vaccination was considered. In the OWSA, the ICER was highest (MXN 489,004) for the analysis of vaccine effectiveness. In the PSA, compared to no vaccine, the maternal RSV vaccine remained the cost-effective option in the large majority (64–93%) of simulations of a plausible range of WTP per QALY gained.

Several economic evaluations have been published on the cost-effectiveness of maternal vaccination with RSVpreF. Consistent with the findings in this study, compared to no intervention, the RSVpreF vaccine was associated with better clinical and economic outcomes across different countries [43,44,45]. A recent analysis evaluated the impact of maternal RSV vaccination in the United Kingdom (UK), reporting that seasonal vaccination could be cost-effective at a cost of up to £80 per vaccination, with a WTP threshold of £20,000 per QALY. The study found that vaccinating 60% of pregnant women throughout the year could result in a 32% reduction in hospitalizations. In this report focusing on the Mexico setting, a base case with 60% vaccine uptake estimated that hospitalizations would decrease by 33%, which is consistent with the results from the UK study. A study by Shoukat et al. (2024) reported that RSVpreF vaccination is considered cost-effective in Canada when the vaccination costs are ≤ CAD 160 from a payer perspective (with a WTP threshold of CAD 50,000 per QALY gained) [45].

This study has limitations for consideration when understanding the results. First, direct effects on vaccinated pregnant people, and potential effects for upper respiratory tract infections, disease transmission, secondary infections, and long-term sequelae of RSV-LRTI in infants, were not taken into account. Second, adverse events associated with the maternal RSVpreF vaccine were not included in the analysis. However, given their rarity and largely mild nature, adverse events are likely to have minor impact on results. Third, due to a lack of Mexico-specific values, data obtained from other countries in South America were utilized for a few model parameters (e.g., caregiver work loss days, travel cost per RSV episode, and distribution of vaccine uptake by wGA). To account for the uncertainties regarding these parameters, scenario and sensitivity analyses were conducted. Fourth, due to limited data availability for hospitalization-, ED-, and outpatient-specific visit costs stratified by status of gestational age at birth, these costs were assumed to be uniformly distributed. Fifth, as noted by authors in the publication by Mata-Moreno et al. (2024) [15], the number of fatalities in their study was relatively small; thus, there was a potential underestimation of the case fatality rate used in the current base case analysis. To mitigate the uncertainty concerning this parameter, a scenario analysis was conducted utilizing alternative case fatality rates. Finally, the indirect impact of maternal RSV vaccination among other populations such as the elderly and others residing in a household with infants was not considered in this study. Further studies could consider this to understand both the economic and clinical outcomes of the RSVpreF vaccine more comprehensively.

## 5. Conclusions

The findings in this study may inform future policy discussions about the inclusion of the maternal RSVpreF vaccine in clinical practice in Mexico. This cost-effectiveness analysis showed that the maternal RSVpreF vaccine is likely cost-effective and potentially cost-saving compared to no vaccination from both the Mexican healthcare system and societal perspectives. Maternal RSVpreF vaccination has the potential to avoid a high number of hospitalizations, emergency room visits, and outpatient visits and would likely be a cost-saving program in Mexico.

## Figures and Tables

**Figure 1 vaccines-13-00077-f001:**
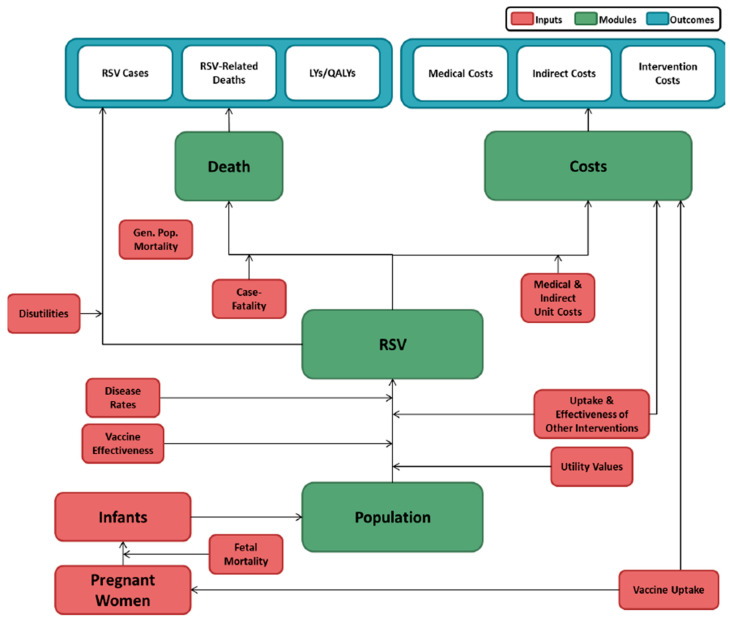
Model schematic.

**Figure 2 vaccines-13-00077-f002:**
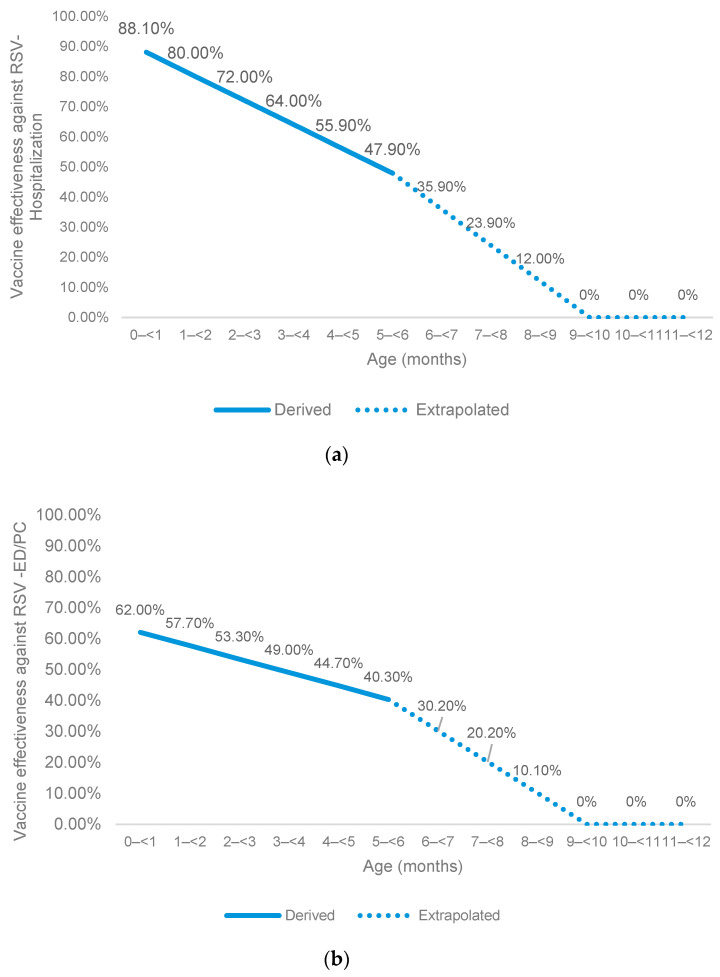
Vaccine effectiveness to protect against (**a**) RSV-LRTI hospitalization and (**b**) RSV-LRTI treated in ED or PC. This was applied to full and late preterm infants born at least 2 weeks after the mother was administered the maternal vaccine. ED: emergency department; PC: primary care; RSV: respiratory syncytial virus.

**Figure 3 vaccines-13-00077-f003:**
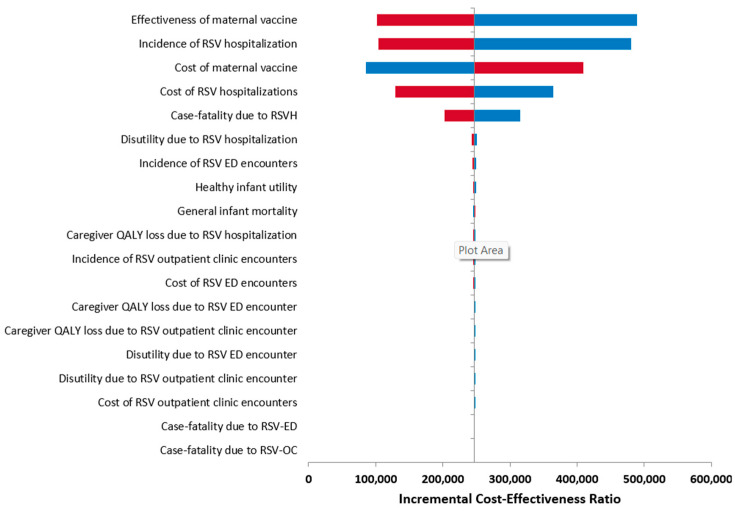
Tornado diagram for the one-way sensitivity analysis (upper [red] and lower [blue] bounds).

**Figure 4 vaccines-13-00077-f004:**
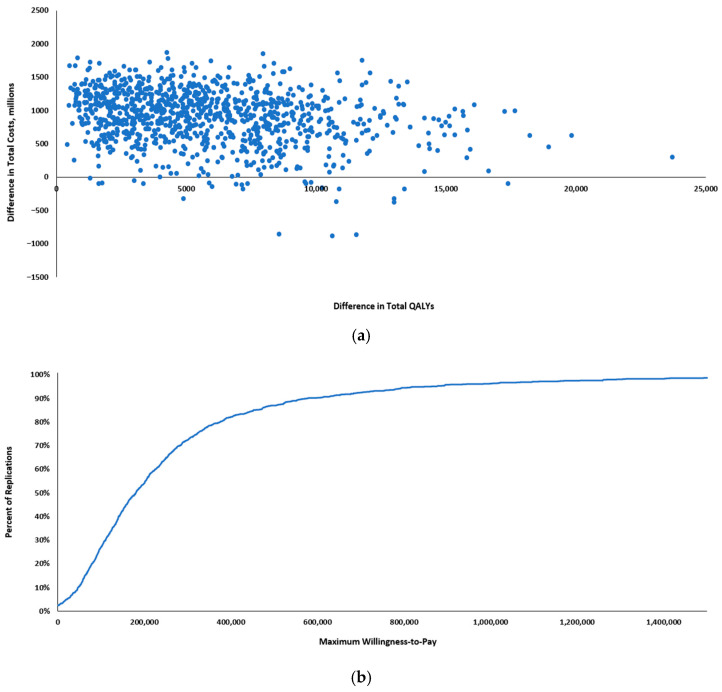
Probabilistic sensitivity analysis results: (**a**) cost-effectiveness plane scatterplot; (**b**) cost-effectiveness acceptability curve. QALYs: quality-adjusted life years.

**Table 1 vaccines-13-00077-t001:** Rates of RSV encounters (per 1000), by age (months).

Infant Age (Months)	Rates of RSV Encounters (per 1000)		
	Hospitalization	Emergency Department	Outpatient
<1	34.0	16.7	16.7
1–<2	60.1	29.5	29.5
2–<3	43.1	21.2	21.2
3–<4	30.2	14.8	14.8
4–<5	26.3	12.9	12.9
5–<6	21.1	10.3	10.3
6–<7	18.6	9.1	9.1
7–<8	15.5	7.6	7.6
8–<9	14.1	6.9	6.9
9–<10	16.2	8.0	8.0
10–<11	11.6	5.7	5.7
11–<12	11.6	5.7	5.7

RSV: respiratory syncytial virus.

**Table 2 vaccines-13-00077-t002:** Base case analysis results.

	Maternal Vaccine	No Intervention	∆
Clinical outcomes (events)			
RSV hospitalization	32,437	48,206	−15,768
RSV ED encounter	18,166	23,671	−5505
RSV outpatient encounter	18,166	23,671	−5505
No. of RSV-related deaths	383	553	−171
Life years (discounted)	38,060,131	38,056,730	3402
QALYs (discounted)	35,700,767	35,697,274	3493
Caregivers’ QALYs lost (discounted)	437	610	−173
Economic outcomes (MXN M)			
Medical care	3131.90	4885.87	−1753.96
Maternal vaccination *	2659.80	0	2659.80
Total	5791.71	4885.87	905.84
ICER			
Cost per LY			266,299
Cost per QALY			247,102

ED: emergency department; ICER: incremental cost-effectiveness ratio; LY: life years; M: million; QALYs: quality-adjusted life years; RSV: respiratory syncytial virus. * Including a vaccine administration cost of MXN 255.49/dose and assuming a cost-effectiveness threshold of 1×GDPpc per QALY gained and the corresponding estimated value of MXN 2105 per dose of RSVpreF. The cost per RSVpreF dose in Mexico is unknown; hence, assumption needed for exploratory purposes of this analysis.

**Table 3 vaccines-13-00077-t003:** One-way sensitivity analysis results.

Parameter	Percent over Base Case	ICER (MXN per QALY)
Effectiveness of maternal vaccine	+25%	101,969
	−25%	489,004
Incidence of RSV hospitalization	+25%	104,380
	−25%	480,282
Incidence of RSV ED encounters	+25%	244,423
	−25%	249,805
Incidence of RSV outpatient encounters	+25%	245,528
	−25%	248,690
Cost of RSV maternal vaccine	+25%	408,859
	−25%	85,345
Cost of RSV hospitalization	+25%	129,458
	−25%	364,745
Cost of RSV ED encounters	+25%	245,561
	−25%	248,642
Cost of RSV outpatient encounters	+25%	246,671
	−25%	247,532
General infant mortality	+25%	249,014
	−25%	245,202
Case fatality due to RSV hospitalization	+25%	203,040
	−25%	315,627
Case fatality due to RSV ED encounters	+25%	247,102
	−25%	
Case fatality due to RSV outpatient encounters	+25%	247,102
	−25%	
Healthy infant utility	+25%	244,961
	−25%	249,280
Disutility due to RSV hospitalization	+25%	243,137
	−25%	251,197
Disutility due to RSV-ED encounters	+25%	246,550
	−25%	247,655
Disutility due to RSV outpatient encounters	+25%	246,550
	−25%	247,655
Caregiver QALY loss due to RSV hospitalization	+25%	245,401
	−25%	248,826
Caregiver QALY loss due to RSV ED encounters	+25%	246,505
	−25%	247,701
Caregiver QALY loss due to RSV outpatient encounters	+25%	246,505
	−25%	247,701

ED: emergency department; ICER: incremental cost-effectiveness ratio; QALY: quality-adjusted life year; RSV: respiratory syncytial virus.

**Table 4 vaccines-13-00077-t004:** Scenario analysis results.

Incremental Outcomes *	Scenario 1: 3% Discount for Costs; 0% for Outcomes	Scenario 2: 7% Discount for Both Costs and Outcomes	Scenario 3: Societal Perspective	Scenario 4: Seasonal Vaccination Strategy	Scenario 5: 40% Vaccine Uptake	Scenario 6: 80% Vaccine Uptake	Scenario 7: Lower Case Fatality Rate	Scenario 8: Higher Case Fatality Rate
Total costs discounted (MXN M)	889.97	921.19	803.88	287.40	603.89	1207.79	905.70	906.00
QALYs (discounted)	12,829	2837	3666	2724	2444	4888	2330	5085
ICER (cost per QALY)	69,375	324,705	219,287	105,529	247,102	247,102	388,725	178,192
% of 1×GDPpc/QALY gained	28.1	131.3	88.7	42.7	99.9	99.9	157.2	72.1

ICER: incremental cost-effectiveness ratio; GDP_pc_: gross domestic product per capita; QALY: quality-adjusted life year. ***** Outcome results are represented as RSVpreF—no intervention strategy.

## Data Availability

The data presented in this study are available upon request from the corresponding author.

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
