# Peer review of "Economic and Clinical Benefits of Bivalent Respiratory Syncytial Virus Prefusion F (RSVpreF) Maternal Vaccine for Prevention of RSV in Infants: A Cost-Effectiveness Analysis for Mexico"

_vaccines, 2025, doi:10.3390/vaccines13010077_

Round 1
Reviewer 1 Report
Comments and Suggestions for Authors
In their scientific study, the authors estimated the cost-effectiveness of using the bivalent RSVpreF vaccine in Mexico. Thus, based on the clinical and economic analysis of the use of RSVpreF vaccine conducted, the authors demonstrated that year-round maternal RSVpreF vaccination will significantly reduce the clinical and economic burden of RSV among infants in Mexico and is likely to become a cost-effective program. The results of this study may be useful in determining a strategy for RSV prevention by health authorities, especially in low- and middle-income countries. Overall, I think the results of this study are positive, but I have a few comments:
(1) It is advisable to digitize the lines of text in the sample article for easy reading by reviewers and editors.
(2) Indicate which two antigens the bivalent RSVpreF vaccine is directed against (RSV A and B). Maybe we are talking about Pfizer's RSVpreF vaccine?
(3) At the same time, it should be noted that according to the results of the meta-analysis, the bivalent RSVpreF vaccine did not prevent the development of RSV infections in children, but the probability of developing a severe form of the disease was significantly lower (PMID: 37018474).
Reviewer 2 Report
Comments and Suggestions for Authors
REVIEW
for the manuscript “Economic and Clinical Benefits of Bivalent Respiratory Syncytial Virus Prefusion F (RSVpreF) Maternal Vaccine for Prevention of RSV in Infants: A Cost-Effectiveness Analysis for Mexico”
Authors: José Luis Huerta, Robyn Kendall, Luka Ivkovic, Carlos Molina, Amy W. Law and Diana Mendes
The title of the article accurately reflects its content.
A brief summary
Aim of the paper was estimation of the clinical and economic burden of RSV infection
among infants in Mexico, with and without a year-round RSVpreF maternal vaccination program.
In this study, data on the expected economic effect from the application of bivalent respiratory syncytial virus prefusion F (RSVpreF) maternal vaccine for prevention of RSV infection in infants for Mexico presented.
Contribution: Specific models of the cost-effectiveness analysis were applied. The authors showed that a year-round RSVpreF vaccine administered to 1,891M pregnant women would prevent 15,768 hospitalizations, 5,505 emergency department cases, and 5,505 physician office visits annually, averting MXN$1,754M in direct medical costs with an increase of 3,402 life years.
Strengths: By extrapolating data from other countries on vaccine effectiveness and national data on RSV prevalence in Mexico, the authors determined that that vaccinating 60% of pregnant women throughout the year would result in a 33% reduction in hospitalizations. The expected direct medical costs decrease was estimated.
Highlighting areas of weakness:
Introduction. No comments.
Materials and methods.
• “A population-based Markov cohort model was used” - the reference for Markov cohort model must be provided.
• The methods of statistical processing of the research results in the presented article are not specified.
• It is not clear, how quality-adjusted life years were determined?
Methodological inaccuracies: not identified;
Missing controls: not identified;
Specific comments
Inaccuracies within the text or sentences that are unclear:
· Table 1: It is not clear why the indicators emergency department cases and physician office visits annually are equal (5,505)? Did every visit to the doctor lead to the need to contact the emergency department?
· Abbreviations in the text must be explained at first mention (such as MXN$ in the Abstract).
· All typos such as US$$3,677.91 in Introduction must be removed.
Discussion
Was the mother's level of antibodies to RSV determined one month after RSVpreF vaccine implementation and at the birth of the child? In my opinion, it would be useful to mention this data in the Discussion with the relevant references.
In the limitations, the authors rightly point out, that the current study has number of limitations and further studies are necessary to understand both economic and clinical outcomes of RSVpreF vaccine more comprehensively.
References 10-14, 32-34 are given in Spanish. To my mind an English version is needed also. Ref. 22 is not available to a wide range of readers
The scientific content:
• The manuscript is clear, relevant for the field and presented in a well-structured manner; The authors declare in the conclusion that:
• Findings in this study may inform future policy discussions about the inclusion of the maternal RSVpreF vaccine in clinical practice in Mexico.
• This cost-effectiveness analysis showed that the maternal RSV vaccine is likely cost-effective and potentially cost saving compared to no vaccination from both the Mexican healthcare system and societal perspectives.
• The cited references are mostly relevant, however references #10-14 and # 32-34 are given in Spanish. To my mind an English version is needed also.
• Ref. 22 is not available to a wide range of readers.
• Overall 8 (18%) of the cited 44 publications are beyond the last 5 years;
• It does not include an excessive number of self-citations;
The experimental design appropriate to test the hypothesis and, to my opinion, the manuscript can be regarded as scientifically sound. The figures/tables are appropriate and properly show the data.
• Statistical analysis methods were not described.
• The conclusion fully reflects the content of the work.
Rating the Manuscript
To my opinion, the manuscript is original and well-defined. The results provide an advancement of the current knowledge.
• Scope: The work fits the journal scope.
• Significance: The results mostly interpreted appropriately.
• Quality: The article is written in an appropriate way. The data and analyses presented appropriately.
• Scientific Soundness: The study designed correctly and technically sound. The methods, tools, software, and reagents described without sufficient details to allow another researcher to reproduce the results.
• Interest to the Readers: Article may be of interest for the readership of the journal.
• Overall Merit: The work expands current knowledge on vaccine prevention of RSV infection in young children. Maternal RSV vaccination has the potential to avoid a high number of hospitalizations, emergency room visits, and outpatient visits of children affected by RSV and would likely be a cost-saving program.
English Level: The English language appropriate and understandable.
Ethical consideration: The study reported was carried out in accordance with generally accepted ethical research standards.
Overall Recommendation
The paper can be accepted. Authors are given five days for minor revisions.
Reviewer 3 Report
Comments and Suggestions for Authors
Dear Authors,
Thank you very much for your submitted manuscript. Please pay attention to the following comments and questions, pertaining to your manuscript:
1. Introduction. Please provide a brief description of the approved vaccine and the maternal vaccination program: clinical implication, gestational age of vaccination, efficiency and safety.
2. Materials and Methods. Infants without intervention (maternal vaccination) are the controls of your study. Has any other type of active or passive immunization against RSV been excluded during pregnancy and beyond (natural maternal/infant infection, passive immunization during lactation)?
3. Materials and Methods. Was there any serological evidence that all the intervention-group infants were immune against RSV after maternal vaccination?
4. Material and Methods. Have you used the same cohort for the cost/effectiveness and the efficiency/safety study?
5. Results. Can you provide any data about the protection against ICU-admission and mechanical ventilation (invasive/non-invasive) due to severe RSV infection?
6. Results. Is there any calculation of the costs which are related to the treatment of RSV vaccination-related fetal/infant complications?
Best Regards
Round 2
Reviewer 3 Report
Comments and Suggestions for Authors
Dear Authors,
thank you for providing comprehensive and convincing answers to the questions and queries expressed by me and the other Reviewers and made changes, that have contributed to the optimization of your manuscript and increased the publishing potential of your work. I have no further questions and queries, pertaining to your manuscript.
Best Regards